# Compact high power, medium energy electron accelerator for treatment of per- and polyfluoroalkyl contamination in water

Tasha Spohr [1,2]*, Benat Alberdi Esuain[1], Marc Dirsat[1],
Sven Lederer[1], Thorsten Kamps[1,2]

1 Helmholtz-Zentrum Berlin, Berlin, Germany, 2 Humboldt-Universität zu Berlin, Berlin, Germany

☉ These authors contributed equally to this work.
* tasha.spohr@helmholtz-berlin.de

## Abstract

Electron beam water treatment (EBWT) is a promising approach for remediating water contaminated with per- and polyfluoroalkyl substances (PFAS). In this study, we assess the feasibility of using a compact, high-average-power superconducting radio-frequency (SRF) photoinjector as a source for delivering the electron beam parameters required to initiate PFAS degradation. Our goals are twofold: first, to determine whether such a system can achieve the necessary dose and dose rate through sufficient beam energy and power; and second, to establish an experimental platform for investigating how different beam conditions affect degradation pathways. We envision a compact and mobile SRF-based accelerator that can be deployed at contamination hotspots - such as the former Berlin airport Tegel - offering significantly faster and potentially more effective treatment than conventional remediation methods. Based on theoretical analysis and computational modeling, we identify the SRF photoinjector at Helmholtz-Zentrum Berlin (HZB) as a suitable R&D platform. To support experimental validation, we developed a proof-of-concept in-air beamline optimized for balancing dose deposition and thermal management. This setup will enable the systematic study of key operational parameters, including dose rate, energy deposition, and thermal stability, under controlled beam conditions.

## Introduction

Access to safe and clean drinking water is a human right. One threat to this is the high level of contamination by per- and polyfluoroalkyl substances (PFAS), known as "forever chemicals," which are used in products that require water, stain, or grease resistance, such as firefighting foam or rain jackets [1]. Numerous studies have revealed the significant health risks associated with exposure to PFAS, particularly chemicals known as PFOA and PFOS, in both humans and animals [2,3]. These risks include a higher likelihood of cancer or diabetes due to the toxicity of PFAS and their resistance to natural degradation processes. Forever chemicals are now

**Data availability statement:** All relevant data, as well as the FLUKA simulation file used in this study are available in an open-access repository. An ANSYS Workbench project file (.wbpj) is also uploaded, which can be used for thermal and structural analysis. These files can be accessed at Zenodo via the following DOI: https://doi.org/10.5281/zenodo.16894583.

**Funding:** This study was supported by a grant from Hi-Acts (No grand number. This was funded by a Helmholtz-Association Grand.). TS and TK received funding from Hi-Acts for this study. The full name of the commercial company that funded this research is "Helmholtz Innovation Platform for Accelerator-based Technologies and Solutions". For more information, please visit the sponsor's website at https://www.hi-acts.de/de. The funders had no role in study design, data collection and analysis, decision to publish, or preparation of the manuscript.

**Competing interests:** I have read the journal's policy and the authors of this manuscript have the following competing interests: TS and TK received funding from Hi-Acts for this project. Hi-Acts had no role in study design, data collection and analysis, decision to publish, or preparation of the manuscript.

found in groundwater, tap water, bottled water, soil, and even in the blood of the population [4].

Currently, various methods to remediate PFAS contamination in water are being investigated [5]. Established methods like granular activated carbon (GAC) [6], ion exchange (IX), and reverse osmosis (RO) are widely used, generally scalable and regulatorily accepted, but merely filter out the PFAS from water resulting in secondary waste streams (e.g., spent media, brine, or membrane reject) that require high-temperature incineration or secure landfilling - both costly and environmentally problematic [7]. Conventional techniques also offer only limited efficiency for short-chain PFOA compounds removal [8]. Emerging technologies for PFAS removal [9, 10], such as electrochemical oxidation by UV light-driven photocatalysis [11] and non-thermal plasma degradation, offer partial degradation of PFAS, especially the short-chain variants, but often suffer from low energy efficiency, limited scalability or acidic water conditions after the treatment [12]. In contrast to these, electron beam water treatment (EWBT) is a destructive, additive-free method that directly cleaves PFAS molecules through radiolysis [13]. EBWT exhibits high potential due to its effective-ness across a broad PFAS spectrum, minimal waste generation, and compatibility with compact, energy-efficient accelerator designs, making it a strong candidate for future sustainable remediation solutions. An electron beam with tailored energy and average power can clean wastewater from industrial processes and degrade PFAS in water. By injecting an electron beam from an accelerator with energies up to 10 MeV into the contaminated water, $H_2O$ molecules break down into free radical electrons, among other products (see Fig 1). These electrons then initiate the degradation of PFAS into shorter chains or biodegradable molecules [11].

EBWT technology has demonstrated, in testing sites, the ability to achieve over 90% removal of perfluorooctanoic acid [13] and exhibits high defluorination efficiency [14]. Total doses ranging from 50 kGy to 1000 kGy have been shown to effectively remove PFOA and PFOS, with PFOA generally being more susceptible to breakdown [15]. EBWT can also degrade PFAS precursors; however, short-chain PFAS exhibit greater resistance to defluorination [16] and likely require higher dose levels. This broad range of dose levels strongly underscores the need for more quantitative experiments to determine the optimal dose level for the specific target pollutants.

**A case study: PFAS contamination at TXL airport in Berlin.** In the framework of this project, we developed a case study addressing a concrete PFAS contamina-tion in Berlin, serving as an example for similar problem cases worldwide. The Berlin metropolitan area is home to four million people and numerous industrial complexes. At the site of the former military section of Tegel airport (TXL), past fire training activ-ities have resulted in significant groundwater contamination with PFAS [17]. A water quality report from Berliner Wasserbetriebe (the municipal water service) revealed high concentrations of PFAS at the Tegel waterworks, of 18 ng L$^{-1}$ for the PFAS-4 and 34 ng L$^{-1}$ for the PFAS-20 groups [18]. Under the new EU Drinking Water Direc-tive (2020/2184), to be implemented in 2028, a combined limit of 100 ng L$^{-1}$ applies to the PFAS-20 group, comprising 20 specified perfluorocarbon and perfluorosulfonic acids (C4–C13) [19]. For PFAS-4, comprising of PFOA, PFNA, PFHxS, and

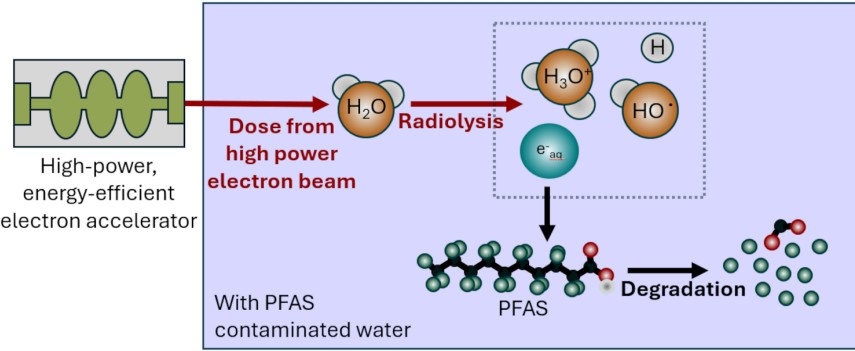

**Fig 1. Principle of PFAS treatment using EBWT with accelerator beams.** The electron beam irradiates the water and starts the radiolysis process. The created hydrated electrons can then react with the PFAS in the water and start the degradation process.

PFOS, this limit will be at $20\,ng\,L^{-1}$ [20]. A report to the Berlin parliament indicates that 29 out of the 131 wells at the Tegel waterworks close to TXL exceed the 2028 PFAS limits, with three wells taken out of operation to prevent pumping highly contaminated water [21]. Water from these wells has been treated since 2022 using a PFAS purification system, which includes two parallel filtration lines equipped with four activated carbon (GAC) filters. The treated water is reintroduced into the water system at potable water quality levels. The activated carbon filters require replacement approximately once or twice a year [17]. The purification facility was built after two years of planning, at an initial cost of approximately EUR 2.5M, with annual operating expenses of approximately EUR 0.86M. Operating costs are primarily driven by energy consumption, iron sludge removal, and the biannual replacement of GAC filters. The expected operation time is 15 years [21]. The scenario at TXL in Berlin is common for airports, airfields, chemical and mineral oil refining plants, and past fire training sites. The average volume throughput achieved in 15 GAC sites across Germany in the years from 2009 to 2014 is $25\,m^3\,h^{-1}$ for water with $1\,\mu g\,L^{-1}$ to $100\,\mu g\,L^{-1}$ PFAS contamination [22].

Our vision is to develop an alternative approach with accelerator-driven EBWT delivering an efficient and sustainable solution to the problem. The EBWT accelerator would be compact and movable, occupying the space of two shipping containers, and could be moved to PFAS contamination sites. The treatment time should be significantly reduced with a throughput of more than $1000\,m^3\,d^{-1}$ for water with contamination levels of $100\,\mu g\,L^{-1}$. With a small fleet of EBWT accelerators, several contamination sites can be supported per year.

**Electron accelerators for EBWT.** As particle accelerator technology continues to advance, become more sustainable, and become easier to operate, employing electron accelerators for EBWT has become increasingly appealing. Since electron energy is the key factor in determining the penetration depth of the accelerated electron beam into water or sludge, medium energies (1 MeV < $E$ < 10 MeV) are necessary for achieving high penetration depth, allowing for high water throughput. Higher beam energies than 10 MeV result in neutron production via bremsstrahlung, leading to activation, which should be avoided. The demand for high power and energy in EBWT systems necessitates efficient, compact, and cost-effective accelerators. Various technologies are available for accelerating electrons to beam energies between 1 MeV to 10 MeV, each offering distinct advantages in terms of efficiency, footprint, and operational complexity. Normal-conducting radio-frequency (NCRF) accelerators are widely utilized in industrial applications and can supply 10 MeV beam energy in compact setups. The technology is limited in scaling up for EBWT levels of operation due to thermal management issues at high average power, limited flexibility in beam parameter patterns, and inadequate energy efficiency.

Most test installations of accelerator-driven EBWT rely on variants of electrostatic high-voltage DC acceleration. An example is the Korean pilot wastewater treatment plant at the Daegu Dyeing Industrial complex. There, a 400 kW (with 1 MeV beam energy and 400 mA average current) electron beam accelerator [23,24] is deployed to remove dye and

organic compounds from 10,000 $m^3$ of wastewater daily [25]. The capital cost of the accelerator plant is USD 4M, the treatment cost is 0.3 USD $m^{-3}$ $kGy^{-1}$. At full capacity of 10,000 $m^3$ per day this results in USD 1M per year operation cost (with depreciation and interest) [26]. The maximum energy of this type of accelerator is limited by the achievable voltage and acceleration gap. Other technologies, allowing for higher beam energy, could enable deeper penetration into the water, thereby allowing for the treatment of more water or denser materials, such as sludge.

Another interesting option to reach 10 MeV in a compact setup is the Rhodotron [27]. A Rhodotron is a high-power electron accelerator based on a modified cyclotron-like architecture. Unlike a conventional cyclotron (which uses a static magnetic field and a spiral trajectory), the Rhodotron uses a coaxial NCRF cavity as an accelerating gap, through which electrons are recirculated multiple times via magnetic arcs. Each pass of the NCRF coaxial cavity adds energy to the beam as it crosses the accelerating gap. This design enables continuous wave acceleration to typical beam energies of 1 MeV to 10 MeV, with beam powers ranging from 10 kW to nearly 1 MW. The key components of the Rhodotron include dipole magnets arranged around the RF cavity, an electron gun, and external beam lines [28]. The capital cost of an industrial Rhodotron for 700 kW beam power and up to 7 MeV beam energy is USD 10M (extrapolated from USD 7M in 2014 price level [29]). The operation costs are on the order of USD 5M per year [30]. A Rhodotron could be a promising option for a fixed installation at a larger site; however, it does not yet meet the specifications for a cost-effective and compact movable, energy-efficient, and sustainable EBWT accelerator.

A promising alternative approach involves compact superconducting radio-frequency (SRF) accelerators, which offer greater efficiency in generating and accelerating high-power beams compared to conventional systems [31]. This technology enables exposure to higher, tailored radiation doses, potentially providing an effective solution for breaking down many persistent PFAS compounds in wastewater treatment. Dhuley et al developed a design study for a 1 MW average beam power with 10 MeV beam energy SRF accelerator for PFAS treatment at Fermi National Accelerator Laboratory (FNAL) in the US [32]. A water cleaning plant based on this accelerator should be able to support the Metropolitan Water Reclamation District in Chicago, USA, and clean up 8000 $m^3$ $d^{-1}$ of waste activated sludge. The cost analysis for this study revealed that such an EBWT accelerator would have a capital cost of approximately USD 8M and treatment costs of 0.14 USD $m^{-3}$ $kGy^{-1}$. While the initial investment is twice as high as that of the Daegu pilot plant, the operating costs are half as much. At Thomas Jefferson National Accelerator Laboratory (TJNAF), a 1 MeV beam energy with 1 MW average beam power SRF accelerator was designed by Ciovati et al in an attempt to treat flue gases and wastewater [33]. The total cost of such an accelerator, able to flue gases at a rate of 10,000 $m^3$ $d^{-1}$, comes to USD 4.5M. While these costs are significant, research can be conducted to reduce the investment and operational expenses of the SRF accelerators. Given the compact dimensions of the FNAL proposed accelerator of 4 m × 2 m × 2 m, including the cryosystem and RF coupling systems, this machine would be transportable to heavily contaminated sites or industrial complexes to treat water where it is most toxic. Therefore, SRF accelerators present a promising approach for EBWT, particularly for the treatment of PFAS. Even with a relatively compact size, they can generate electron bunches with sufficiently high energies and considerable power, which is the most crucial aspect in the degradation of PFAS. They are not significantly more expensive and can deliver higher doses than alternative solutions.

For our case study at TXL airport, the specification for a compact and movable SRF accelerator serving PFAS hotspots would be capital costs of less than EUR 5M for the SRF accelerator plus required infrastructure and operation costs lower than EUR 0.5M per year. It should deliver enough beam power at several MeV beam energy to be able to allow a throughput of more than 1000 $m^3$ $d^{-1}$ for water with contamination levels of 100 μg $L^{-1}$ at treatment costs of less than 0.10 EUR $m^{-3}$ $kGy^{-1}$. The primary goal of this study is to develop an R&D platform to determine the optimal dose rate, beam power, and beam energy, as well as the trade-off between these parameters, suitable for driving specific PFAS degradation pathways. The SRF photoinjector at SEALab can serve as this platform, as it offers versatile performance for operation modes ranging from low-power proof-of-concept experiments to high-power close-to-real tests. This research must be aligned with developments on SRF accelerator facilities, outside the scope of this study, to make these accelerators even more energy-efficient and compact [34].

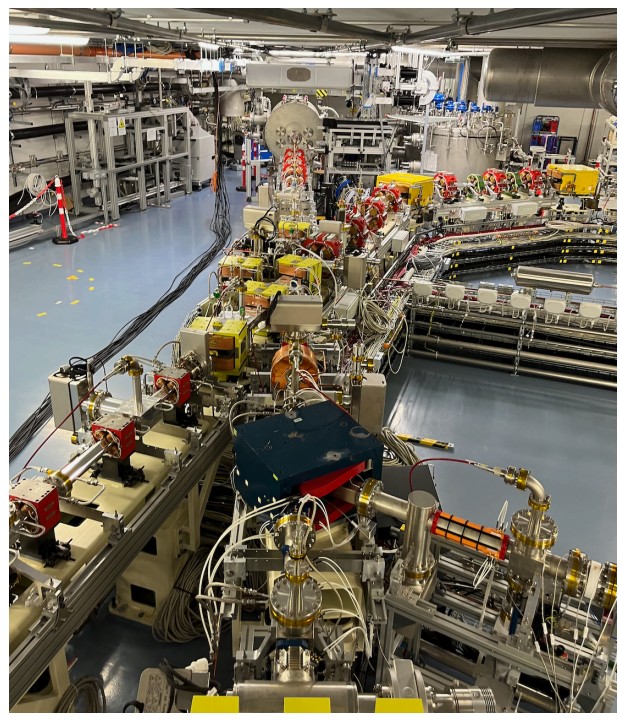

**Fig 2**. **Photograph of the SEALab accelerator platform.** Shown is the SRF photoinjector and diagnostic beamline. The straight section to the left leads to the experimental site.

## The SRF photoinjector at SEALab

SRF accelerators present an excellent option for optimizing the treatment of contaminated water. Helmholtz-Zentrum Berlin (HZB) is engaged in a robust accelerator R&D program to develop sustainable accelerator solutions that provide tailored beam properties available for various applications. A key component of R&D is the SEALab accelerator facility, centered around an SRF photoinjector [35]. We initiate our approach at the SEALab SRF photoinjector (Fig 2) as an R&D platform to identify the dose and beam parameters necessary for efficient treatment and to advance technological innovations that support the sustainability of a compact EBWT SRF accelerator.

The SEALab SRF photoinjector is an accelerator designed initially to serve as an injector for an energy-recovery linac (ERL). It was designed to generate and accelerate electron bunches up to 2.5 MeV to 6.5 MeV beam energy with a high average current of up to 100 mA, resulting in a total beam power up to 650 kW (see Table 1). The high power level and exceptional versatility in beam parameters (energy, energy spread, bunch pattern, beam size) make it an ideal R&D platform for EBWT targeting PFAS contamination in wastewater or soils.

**Table 1**. **Core parameters for the SRF photoinjector.** Initial SEALab specifications and demonstrated parameters [35], plus specifications for a dedicated EBWT mode and the proof-of-concept phase. Note that the demo parameters for SEALab have been observed with different SRF gun systems.

| Parameter | Symbol | SEALab | | EBWT | |
|---|---|---|---|---|---|
| | | Specs | Demo | Mode | POC |
| Energy | $E$ | 6.5 MeV | 2.5 MeV | 2 - 9.5 MeV | 2 - 6.5 MeV |
| Avg. Current | $I_{avg}$ | 100 mA | 8 µA | ≤ 10 mA | 3 µA |
| Power | $P$ | 650 kW | 20 W | ≤ 95 kW | 6 - 19.5 W |

Unlike the designs from FNAL and TJNAF, SEALab provides the parameter space for testing future mobile treatment facilities, which are small and compact enough to be brought on site. With our planned experiments at SEALab, the exact parameter space for PFAS removal using EBWT can be optimized. In further studies, it can be investigated whether this technology is suitable for compact, movable accelerators. One key benefit of SEALab's setup is that the entire beamline is already in place, requiring minimal adjustments before initiating the first experiments to investigate optimum beam properties for specific PFAS contaminants.

## SRF photoinjector driven EBWT

With the SEALab accelerator already in place, we needed to evaluate its feasibility for treating PFAS-contaminated water. This section presents the physical model for dose generation and delivery, along with the current setup's limitations. It also outlines the following steps to initiate this experiment once the accelerator is fully operational.

Starting with estimates from analytical considerations, we utilized FLUKA [36–40] simulation software to study the interaction of the electron beam with the water samples, compute the dose deposition in a close-to-realistic environment, and prepare for a proof-of-concept experiment. Additionally, Ansys [41] is used to study the thermal behavior and heat dissipation of the electron beam at the various intersections of the experimental setup.

### Physical model for dose generation

To degrade PFAS with EBWT, accelerated electrons must first interact with water to start the radiolysis process. The high-energy electrons break down $H_2O$ molecules into various reactive species, including hydrated electrons ($e^-_{aq}$) [42]. The hydrated electron can react with PFAS, triggering the breakdown of its strong C-F bonds [13]. The PFAS then follow multiple possible reaction pathways and can ultimately form shorter-chain compounds or other chemicals [11].

One of the most critical parameters for the radiolysis is the G-value $G(x)$ of material $x$, which gives the chemical yield as the number of molecules $n(x)$ transformed per energy $E$ imparted to the system [43]:

$$G(x) = \frac{n(x)}{E}.$$ (1)

Previous studies have found that the relative amount of hydrated electrons to scavenging species, such as $H_2$, is crucial for the degradation of PFAS. Scavenging species can react with hydrated electrons before those can react with the PFAS compound, ultimately reducing the available $e^-_{aq}$ necessary for degradation. A high G-value for the hydrated electron and a low G-value for scavenging species increase the degradation rate of PFAS [13].

The radiation dose $D$ and chemical yield $G$ are closely related. The amount $n(x)$ of transformed molecules of species $x$ is:

$$n(x) = m \cdot G(x) \cdot D$$ (2)
$$= m \cdot G(x) \cdot \dot{D} \cdot \Delta t,$$ (3)

where $\dot{D}$ is the dose rate and $\Delta t$ the treatment time. The dose is the absorbed energy $dE$ per mass $dm$:

$$D = \frac{dE_{total}}{dm} = \frac{1}{\rho} \frac{dE_{total}}{dV} = \frac{1}{\rho} \frac{dE_{total}}{dz \cdot dA},$$ (4)

where $\rho$, $V$, and $A$ are the given material's density, volume, and illuminated area. The electron fluence

$$\phi = \frac{dN}{dA} = \frac{1}{e} \cdot \frac{dq_B}{dA}$$ (5)

characterizes the number of electrons $dN = dq_B/e$ per area $dA$, where $e$ is the elementary charge and $q_B$ the total charge of the beam. Using $I = \frac{dq_B}{dt}$ We get the fluence rate:

$$\dot{\phi} = \frac{d\phi}{dt} = \frac{1}{e} \cdot \frac{dq_B}{dt \cdot dA} = \frac{I}{e \cdot dA} \tag{6}$$

The total energy loss of $dN$ electrons traveling through a material of depth $dz$ is:

$$dE_{total} = \left(\frac{dE}{dz}\right) \cdot dz \cdot dN. \tag{7}$$

The mass stopping power $S/\rho$ of a given material gives the energy loss of particles due to interactions with the material:

$$\frac{S}{\rho} = -\frac{1}{\rho}\frac{dE}{dz}. \tag{8}$$

Thus, the dose becomes:

$$D = \frac{dE_{total}}{dm} = \frac{\left(\frac{dE}{dz}\right) \cdot dz \cdot dN}{\rho \cdot dz \cdot dA} = \frac{1}{\rho}\frac{dE}{dz} \cdot \frac{dN}{dA} = \phi \cdot \frac{S}{\rho}. \tag{9}$$

And the dose rate is (written in practical units):

$$\dot{D} = \frac{dD}{dt} = \dot{\phi} \cdot \frac{S}{\rho} \tag{10}$$

$$\dot{D}[\text{Gy s}^{-1}] = \frac{I[\text{nA}]}{A[\text{cm}^2]} \cdot \frac{S}{\rho}\left[\frac{\text{MeV} \cdot \text{cm}^2}{\text{g}}\right] \tag{11}$$

The dose rate is critical in determining the steady-state concentration of hydrated electrons [13]. A higher dose generally correlates with increased production of reactive species, including hydrated electrons. Since a higher dose rate also means an increased production of scavenging species, an optimal dose rate can be found. Our goal is to determine the dose rate using the experiment at SEALab and investigate a possible cut-off dose rate above which the scavenging species dominate the formation of hydrated electrons. Previous studies indicate that dose rates on the order of kGy/s are most effective, achieving the optimal balance between hydrated electrons and scavenging particles [13]. The total dose is the (constant) dose rate multiplied by the treatment time $\Delta t$:

$$D[\text{Gy}] = \frac{I[\text{nA}]}{A[\text{cm}^2]} \cdot \frac{S}{\rho}\left[\frac{\text{MeV} \cdot \text{cm}^2}{\text{g}}\right] \cdot \Delta t[\text{s}]. \tag{12}$$

The total dose and dose rate are both dependent on the electron beam current in the accelerator and the size of the treated area. Using raster beam magnets, the beam area and thus the treatable area can be controlled. The beam current is highly flexible in the SEALab accelerator, with tuning possible for both the frequency $f_{rep}$ and the bunch charge $q_B$. While the stopping power $S$ is inherent to material properties, both the dose rate and total dose can be manipulated by accelerator setup parameters, such as bunch charge, repetition rate, and beam spread area.

## Electron beam delivery

To ensure precise beam delivery, beam dynamics simulations have been used to model the electron beam optical propagation from the source through the electromagnetic lenses up to the exit window. The simulations ensure the beam exits the vacuum through a thin titanium window with defined properties and propagates effectively through the air before reaching the sample holder.

With these considerations, we checked the feasibility of SEALab in implementing an EBWT POC operation mode suitable for parameter exploration in an experiment. We verified that the experimental dose rate, based on SEALab's parameters, would be sufficient to achieve the minimum required total dose for PFAS degradation of at least 10 kGy [15] within a reasonable time frame, considering machine protection considerations. Additionally, the water must remain below boiling temperature to avoid complications such as condensation and pressure issues in the water sample holder. The energy deposited in the exit window must not cause melting or structural failure, even with the pressure differences between the vacuum inside and the air outside.

For initial calculations, we began with a beam operation mode typically used for machine commissioning at an $E = 2.5$ MeV beam energy and $I = 500$ nA average current. Below are the calculations with the final setup's parameters (see Table 2) to test the feasibility of our experiment.

**Power calculation:** The power delivered by the accelerator is calculated via:

$$P[\text{W}] = E[\text{MeV}] \cdot I[\mu\text{A}] \tag{13}$$

$$P_{\text{EBWT}} = 1.25 \, \text{W} \tag{14}$$

Normalized to the volume $V_{\text{Water}} = 9.1$ mL of the treated water one obtains:

$$P_{\text{norm}} = \frac{P}{V} \tag{15}$$

$$P_{\text{EBWT,norm}} = 137.7 \, \text{mW cm}^{-3} \tag{16}$$

**Dose calculations:** Assuming a stopping power of $S = 1.868$ MeV cm$^{-1}$ in water [44], the dose rate is calculated using Eq (11):

$$\dot{D}_{\text{EBWT}} = \frac{500 \, \text{nA}}{9.1 \, \text{cm}^2} \cdot 1.868 \, \frac{\text{MeV} \cdot \text{cm}^2}{\text{g}} \tag{17}$$

$$\dot{D}_{\text{EBWT}} = 102.6 \, \text{Gy s}^{-1} \tag{18}$$

**Temperature considerations:** At steady state, the power input $P$ is equal to the convective heat loss [45]:

$$Q = h \cdot A \cdot (T_s - T_\infty), \tag{19}$$

**Table 2. Sample holder geometry parameters.** Using FLUKA simulation, water depth and water radius were optimized.

| Parameter | Symbol | Value |
|---|---|---|
| Sample Holder Dimensions | w×h×d | (40×40×50) mm$^3$ |
| Water Radius | $R_{\text{wtr}}$ | 17 mm |
| Water Depth | $d_{\text{wtr}}$ | 10 mm |
| Water Volume | $V_{\text{Water}}$ | 9.1 mL |
| Window Thickness | $d_{\text{entr}}$ | 130 µm |

where $Q$ is the heat transfer rate, which we can approximate with $Q = P$. $T_s$ is the steady-state temperature and $T_\infty$ the ambient temperature. $h$ is the convective heat transfer coefficient, assumed to be $5\,\mathrm{W/(m^2\,K)}$, which approximates natural convection in still air. $A$ is the area of the body that is in contact with the heated water and can dissipate heat, so it is the surface area of the sample holder or exit window. Rearranging to solve for $T_s$ for the water sample:

$$T_s = \frac{P}{h \cdot A} + T_\infty \tag{20}$$

$$T_{\mathrm{EBWT,wtr},s} = 44\,°\mathrm{C} \tag{21}$$

And for the exit window:

$$T_s = \frac{P}{h \cdot A} + T_\infty \tag{22}$$

$$T_{\mathrm{EBWT,extWindow},s} = 77\,°\mathrm{C} \tag{23}$$

Compared with previous studies, we achieved total doses in the range of those in successful earlier experiments [16]. While our expected dose rate is by an order of magnitude smaller than that used in previous studies (1.58 - 20 kGy/s [16]), it will be tunable. As Eq (11) shows, the dose rate goes linearly with the beam current, which can be tuned at the SEALab accelerator from nA to mA, allowing for high dose rates. It has also been found that lower dose rates yield better results. Part of the study aims to determine whether this dose rate is sufficient and how to optimize it for better results [13]. Within a realistic time frame, a total dose similar to that of previous successful studies can be achieved. Later, Ansys simulations show that with these parameters, the exit window is neither melted nor directly destroyed by stress or boiling water.

## Setup for EBWT POC experiments

Since the calculations indicate that EBWT for PFAS is possible with the SEALab accelerator in principle, a proof-of-concept experiment can be set up. In this section, we outline the experimental setup and the simulations used to optimize the beam parameters and define an Electron Beam Water Treatment Mode (EBWT Mode).

The SEALab accelerator provides an excellent setup for conducting in-air experiments. Our experiment will be strategically located at the first bend of the beamline, where the setup benefits from a considerable distance from the electron gun and booster, allowing for the implementation of essential vacuum protection systems and a relatively easy path to steer the beam. In the SEALab photoinjector (Fig 3), an electron beam is created and then accelerated by the SRF gun. Electromagnets steer the beam onto the primary beamline, directing it towards the first bend of the recirculation arc. Using more electromagnets, the beam is directed to the experimental site and spread open to maximize the area with which it strikes the vacuum exit window. This is important for the window's longevity and for maximizing the amount of water irradiated in the in-air experiment.

After passing through this exit window, the beam travels through the air before entering the sample holder (Fig 4), which is securely housed in a shielding container to ensure radiation protection and temperature convection. The sample holder is a rectangular box made of stainless steel, chosen to provide shielding for electrons that do not deposit their entire energy in the water, as well as for the Bremsstrahlung photons created. Additionally, it was chosen to dissipate the heat generated by the reactions in the water and the sample holder.

Inside that sample holder, the water samples are stored in cylindrical holders to optimize dose distribution. This design choice is key to maximizing the effectiveness of the Gaussian beam's interaction with the sample. The water will act as a beam dump, and the delivered dose will degrade the PFAS. A small opening at the top of the holder allows water to be added and removed using a pipette, as well as some pressure regulation in case of water vaporization. This setup is

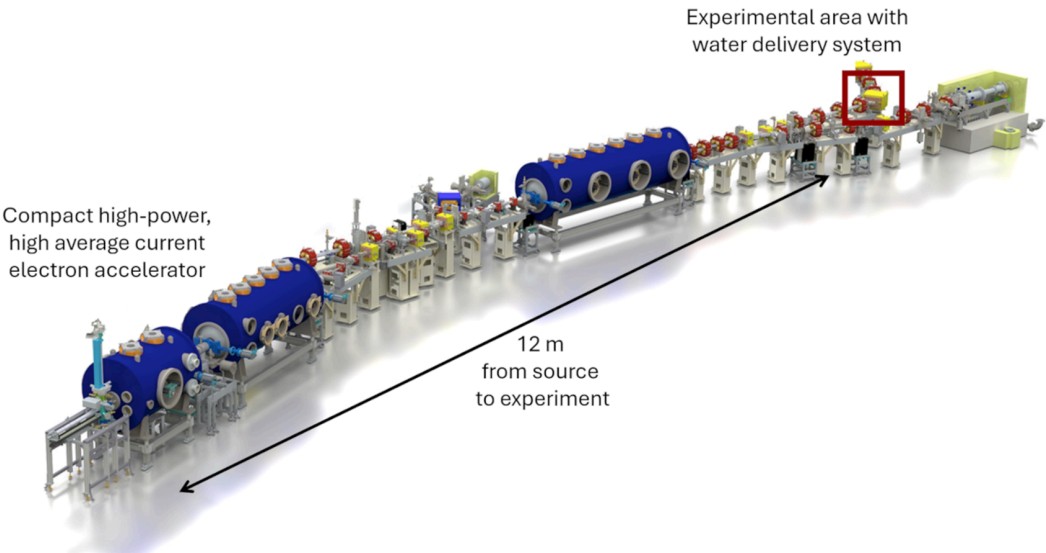

**Fig 3**. **SEALab SRF photoinjector with water delivery system.** The electron gun and booster module accelerate the electron beam up to 6.5 MeV. The electrons can then proceed through the straight section, into the first bend, and reach the water experimental area (marked in red).

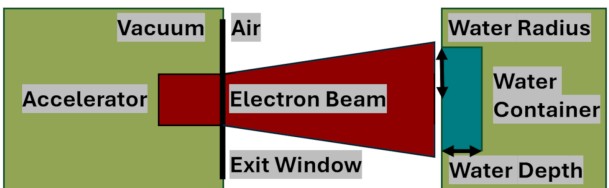

**Fig 4**. **Sketch of the accelerator and experimental setup.** Electrons will be accelerated and leave the beamline through a vacuum exit window. Then, they reach the water container where the PFAS degradation will take place.

conveniently positioned on an existing table in the experimental hall. Additional shielding is installed as needed to protect against scattered electrons and photons. Access heat can be delivered to the surrounding air.

## EBWT operation mode at SEALab

We performed simulation studies to find optimal parameters for the EBWT mode of the SRF photoinjector and the experiment's setup. Fig 5 shows the experimental setup in FLUKA and some dose rate simulations in the water sample.

**Water radius and FWHM.** The radius of the water sample determines how many electrons hit the surface for a specific beam radius. If $R_{wtr} \ll x_{FWHM}/2$, with $x_{FWHM}$ being the full width half maximum (FWHM) in transversal direction, then the electrons cannot irradiate the entire volume, and the average deposited dose reduces. For $R_{wtr} \gg x_{FWHM}$, the electrons will also irradiate the sample holder around the water container; fewer electrons hit the target, and less dose is delivered. As shown in Fig 6, the dose rate delivered reaches a maximum at a specific water radius. Since the water radius significantly influences the water volume, a sufficiently large radius is necessary to irradiate large enough water samples. Various $x_{FWHM}$ values were tested: for small $x_{FWHM}$, the window causes significant beam dispersion. This effect diminishes at around $x_{FWHM} = 3$ mm. The averaged delivered dose rates converge for all $x_{FWHM}$ towards the same curve for high enough $R_{wtr}$. The highest possible $x_{FWHM}$ is determined by the beam pipes diameter, which is $d_{max} = 38$ mm. For a $5\sigma$ beam, this

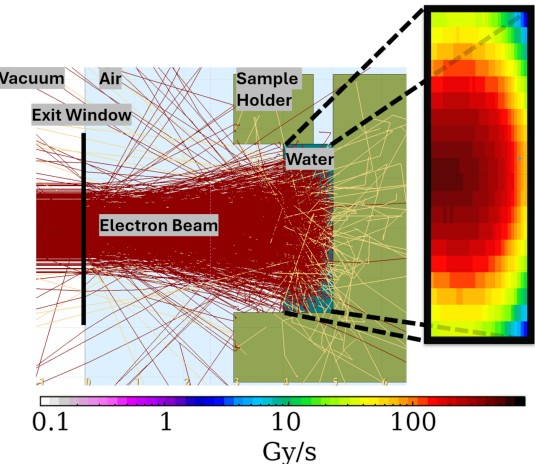

**Fig 5. Experimental SetUp in FLUKA.** The electrons (red) and photons (yellow) are tracked in the x-z-plane from where they exit the vacuum through the exit window and then enter the water holder in the middle of the figure. To the right, the dose rate in the water sample is shown on a logarithmic scale.

means that the maximum FWHM must be:

$$\sigma_{\text{max}} = \frac{38\,\text{mm}}{5} = 7.6\,\text{mm} \tag{24}$$

$$x_{\text{FWHM,max}} = 2\sigma \cdot \sqrt{2\log(2)} = 17.9\,\text{mm} \tag{25}$$

For some extra margin, a maximum $x_{\text{FWHM}} = 17$ mm was assumed. For such a beam configuration, the maximum dose rate is delivered at $R_{\text{wtr}} = 7$ mm (see Fig 6). This radius is still too small to produce a useful water volume. Instead of reaching a maximum of any curve, one can choose a water radius that still corresponds to a high enough dose rate. In Fig 6, this functional region with high enough dose rates and big enough radii is marked in yellow.

Thermal simulations with Ansys were performed to check if differently sized beams might melt or vaporize the exit window or the water. High peak temperatures were observed for small beam diameters, but no melting or vaporization occurred. Any $x_{\text{FWHM}}$ between 2 mm and 17 mm can be used for the setup without damaging it while still reaching high enough dose rates for the shown water radii.

**Depth and energy.** The electron beam loses approximately 2 MeV of energy for every centimeter it travels through water. To ensure effective dose distribution throughout the entire sample, the depth of the water sample is chosen so that the dose reaches the end of the sample holder. This depth is identified as the point at which the dose achieves 50% of its maximum value [46]:

$$d_{50} = E\,[\text{MeV}]/2.33. \tag{26}$$

For a $E = 2.5$ MeV beam, this accounts to 10.7 mm. For simplicity, a depth of $d_{\text{EBWT}} = 10$ mm was chosen. In Fig 7, the dose rate for different water depths was plotted for different beam energies. The yellow area again marks the optimal parameter space. Even for the expected beam energy of 2.5 MeV, a small window of sample size parameters can be found that satisfy these requirements.

With the later Booster update, beam energies will rise to 6.5 MeV. Then, a much higher depth can be chosen, and more water can be irradiated. Table 2 shows the dimensions for the first experimental setup found with these simulations.

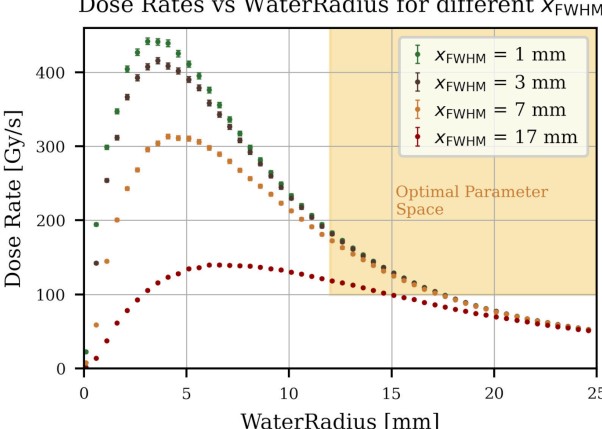

**Fig 6**. **Average dose rate vs water radius.** FLUKA simulation study results. The water radius was increased, and the average dose rate in the water sample was simulated for different $x_{FWHM}$ values. It can be seen that for this specific setup and for useful water radius ranges (larger than 1 cm), the change in $x_{FWHM}$ does not significantly alter the delivered dose.

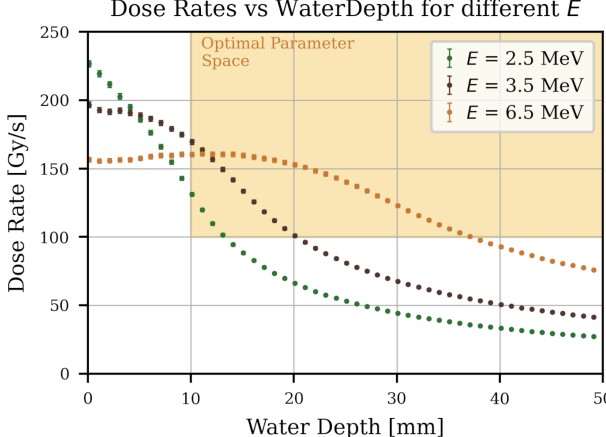

**Fig 7**. **Average dose rate vs water depth of sample.** FLUKA simulation study results. The water depth was increased, and the average dose rate in the water sample was simulated for different energies. Note: This is not the dose rate distribution over the depth of the water sample.

**Current.** The current directly influences the dose rate at the water target, as well as the power experienced at the sample and exit window (Eqs (11) and (13)). Using Ansys simulation and assuming different heat coefficients, a maximum current was found for which the water stays below boiling temperature while still receiving a high enough dose rate. The maximum beam current for a water-cooled (heat transfer coefficient $h = 500 \, W/(m^2 \, K)$) water sample holder and an air-cooled ($h = 100 \, W/(m^2 \, K)$) vacuum exit and water entry window is found to be $I_{max} = 4 \, \mu A$. For some extra margin, a current of $I_{EBWT} = 3 \, \mu A$ will be assumed. This current drastically increases the dose (rate) compared to the previous simulations, which were performed at 500 nA. Table 3 shows the assumed EBWT mode beam parameters.

### Physical setup

After successful convergence tests and analysis of multiple possible setups, the experiment was designed as explained earlier with the parameters found in Table 2. Fig 5 shows the FLUKA modeled version of the setup, with an electron beam

**Table 3**. **EBWT mode beam parameters.** Optimized with FLUKA and Ansys simulations.

| Parameter | Symbol | Value |
|---|---|---|
| Beam Energy | $E$ | 2.5 MeV |
| Beam Current | $I$ | 3 µA |
| Beam FWHM | $x_{FWHM}$ | 7 mm |
| Beam Size | $\sigma_{x,y}$ | 2.97 mm |

of $\sigma_{EBWT} = 2.97$ mm. The dose rate deposited in the water is also shown on a logarithmic scale for the current $I_{EBWT} = 3$ µA. The vacuum exit window in this setup is a 50 µm thick TI-6AL4V window. This window has a diameter of 41.7 mm and is positioned 40 mm in front of the water sample holder. To account for (back)scattering electrons, additional shielding was placed around the water sample. The averaged dose rate in the water sample is 659.90(24) Gy s$^{-1}$, which leads to a treatment time of about 2.5 minutes if 100 kGy are to be reached (see Table 4). Ansys simulation proved that this type of irradiation does not boil the water (the maximum temperature in the water is 50.2 °C).

A major problem is the uniformity of the dose deposited in the water. Fig 8 shows the dose deposited over the depth of a 1 cm thick water sample holder. With higher energy, a larger distance could be uniformly irradiated, thus increasing the overall dose rate in the sample. Fig 9 shows the dose rate at $y, z = 0$ mm over the horizontal position of the water. The uniformity would increase with a smaller water radius. The experiment will show whether this non-uniformity has a significant influence on the overall degradation of PFAS.

**Table 4**. **Simulated values for the experiment using EBWT mode.**

| Parameter | Symbol | Value |
|---|---|---|
| Avg. Dose Rate | $\dot{D}$ | 659.90(24) Gy/s |
| Treatment Time for 100 kGy | $t$ | 2.5 min |
| Max. Temperature | $T_{max}$ | 50.2 °C |
| Avg. Temperature | $T_{avg}$ | 40.8 °C |

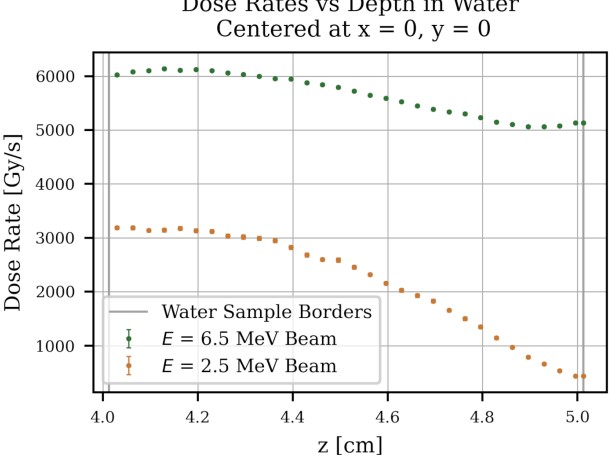

**Fig 8**. **Average dose rate vs depth of water sample.** FLUKA simulation study results. The dose rate is plotted over the depth of the water sample. The dose over depth was simulated for two energy options (2.5 MeV and 6.5 MeV). For the future 6.5 MeV beam, the depth of the water sample can be further increased.

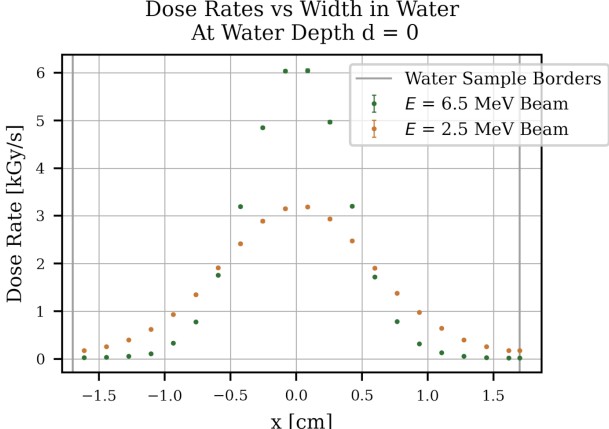

**Fig 9**. **Average dose rate vs horizontal position in water.** FLUKA simulation study results. The average dose rate is plotted along the x-axis of the water sample at the point of entry.

## Accelerator protection measures

The beam vacuum exit window must be designed to withstand the pressure difference and the electron beam power. Furthermore, a beam shutter and machine protection must be installed in case it breaks, preventing particulates from flowing into the accelerator's vacuum system.

**In-air window:** Due to the pressure difference, the Ansys simulation showed that the vacuum exit window deforms up to 0.995 mm. This type of titanium has a tensile yield strength of 845.7 MPa (Ansys database). With the simulations showing a Von-Mises stress of around 530 MPa, the window only deforms elastically. The most important parameters are shown in Table 5.

Further Ansys simulations also show that the temperature in the window does not cause melting, as the temperature does not change significantly for a heat transfer coefficient of $h = 100\,\text{W}/(\text{m}^2\,\text{K})$.

**Fast beam shutter system:** A fast-acting vacuum protection system must be installed to ensure that the accelerator vacuum system is protected from particulate flow in the event that the exit window breaks. In the event of a vacuum leak, we assume any particulates will travel at the speed of sound ($v_s \approx 300\,\text{m/s}$). Then, conservatively estimating a reaction time of 20 ms, the protection system would need to be positioned approximately 6 m from the sensor. This placement is feasible in the experimental hall if the existing shutter is relocated behind the merger of the injection and primary beamlines. Additionally, a small sensor must be installed before the vacuum exit window, enabling the fast shutter to act in time if necessary. This modification can be implemented relatively easily using the existing hardware without compromising the accelerator vacuum system.

**Table 5**. **Simulated values for the vacuum exit window.**

| Parameter | Symbol | Value |
|---|---|---|
| Thickness | $d_{ext}$ | 50 μm |
| Max. Deformation | $\delta_{total}$ | 0.995 mm |
| Max. Stress | $\sigma_{VM}$ | 527 MPa |
| Max. Temperature | $T_{max}$ | 28 °C |

## Conclusion and outlook

The EBWT@SEALab project was initiated to address the pressing challenge of PFAS contamination in water at contamination hotspots. We evaluated the potential of the existing SEALab accelerator for electron beam water treatment (EBWT), examining theoretical performance limits and developing a dedicated operational mode. This effort culminated in the design of a proof-of-concept experiment built around an in-air beamline configuration. The SEALab superconducting radio-frequency (SRF) photoinjector offers a high degree of flexibility in beam parameters. By adjusting the electron beam current and the beam energy, we can systematically vary both the total dose and dose rate. Operation at high current enables dose rates of several kilogray per second (kGy/s), sufficient for initiating PFAS degradation processes. Simulation studies of energy deposition, thermal management, and beam transport formed the final design of the experiment, enabling precise control over key operational parameters, such as dose rate, energy distribution, and thermal stability. These results confirm that the SEALab accelerator provides the beam quality and power required to investigate EBWT under conditions relevant to PFAS remediation.

The next phase of the project will focus on implementing and executing the experiment. Planned activities include the engineering design of a dedicated water treatment platform, the development of a fast shutter system and beam rastering unit, and the establishment of a water analysis network. In collaboration with TU Berlin and Berliner Wasserbetriebe, we will conduct chemical analyses to quantify the degradation efficiency of PFAS across different compound groups. In parallel, we will explore the scalability of the technology for industrial applications and conduct a feasibility study focused on the deployment of compact SRF accelerators for decentralized EBWT.

With beam time scheduled for later this year, we anticipate launching the first in-air experiments to evaluate the effectiveness of electron beam treatment for PFAS-contaminated water.

## Acknowledgments

We would like to acknowledge Carsten Walter and Prashanth Menezes from Technische Universität Berlin for their helpful discussions on the chemistry of PFAS degradation; Holger Huck from HZB for assisting with FLUKA simulations; and Boris Schröder and Sonja Specks from HZB for helping to find industry partners. We would also like to acknowledge Axel Neumann from HZB for supervising the SEALab project and providing helpful feedback on this paper.

## Author contributions

**Conceptualization:** Thorsten Kamps.

**Data curation:** Tasha Spohr.

**Formal analysis:** Tasha Spohr, Sven Lederer, Marc Dirsat.

**Funding acquisition:** Thorsten Kamps.

**Investigation:** Tasha Spohr, Beñat Alberdi Esuain, Sven Lederer.

**Methodology:** Tasha Spohr, Marc Dirsat.

**Project administration:** Thorsten Kamps.

**Resources:** Sven Lederer, Marc Dirsat.

**Software:** Tasha Spohr, Marc Dirsat.

**Supervision:** Thorsten Kamps.

**Validation:** Tasha Spohr, Marc Dirsat.

**Visualization:** Tasha Spohr.

**Writing – original draft:** Tasha Spohr.

**Writing – review & editing:** Thorsten Kamps, Sven Lederer, Marc Dirsat.

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
