## [Decision Letter · Decision Letter 0]

9 Jul 2025

PONE-D-25-19239Compact High Power, Medium Energy Electron Accelerator for Treatment of Per- and Polyfluoroalkyl Contaminations in Water - EBWT@SEALABPLOS ONE

Dear Dr. Spohr,

Thank you for submitting your manuscript to PLOS ONE. After careful consideration, we feel that it has merit but does not fully meet PLOS ONE’s publication criteria as it currently stands. Therefore, we invite you to submit a revised version of the manuscript that addresses the points raised during the review process.

Finding expert reviewers for this manuscript has been challenging. Therefore, since the sole reviewer who accepted and completed the evaluation is an expert in this field, I will exceptionally base my decision on this single review, in line with their recommendation for a major revision.

We look forward to receiving your revised manuscript.

Kind regards,

Alexandre Bonatto

Academic Editor

PLOS ONE

Journal Requirements:

3. Please amend your list of authors on the manuscript to ensure that each author is linked to an affiliation. Authors’ affiliations should reflect the institution where the work was done (if authors moved subsequently, you can also list the new affiliation stating “current affiliation:….” as necessary).’

4. Please ensure that you refer to Figure 7 in your text as, if accepted, production will need this reference to link the reader to the figure.

5. We note you have included a table to which you do not refer in the text of your manuscript. Please ensure that you refer to Table 3, 4, and 5. in your text; if accepted, production will need this reference to link the reader to the Table.

Additional Editor Comments :

Dear Author,

Finding expert reviewers for this manuscript has been challenging. Therefore, since the sole reviewer who accepted and completed the evaluation is an expert in this field, I will adopt their recommendation for a major revision.

Kind regards,

Alexndre Bonatto

Reviewers' comments:

Reviewer's Responses to Questions

**Comments to the Author**

1. Is the manuscript technically sound, and do the data support the conclusions?

Reviewer #1: Yes

2. Has the statistical analysis been performed appropriately and rigorously? 

Reviewer #1: N/A

3. Have the authors made all data underlying the findings in their manuscript fully available?

Reviewer #1: Yes

4. Is the manuscript presented in an intelligible fashion and written in standard English?

Reviewer #1: Yes

5. Review Comments to the Author

Reviewer #1: The current study aims to investigate a proof-of-concept experiment of a new in-air beamline electron beam system to potentially degrade PFAS in aqueous and theoretically other matrices. It is extremely important to investigate innovative technologies to completely degrade PFAS and e-beam technology is one such approach. The authors have done a good job of summarizing existing literature on this field and accordingly comparing their proof-of concept device to these devices

6. PLOS authors have the option to publish the peer review history of their article (what does this mean?). If published, this will include your full peer review and any attached files.

Reviewer #1: No

---

## [Author Response · Author response to Decision Letter 1]

3 Nov 2025

General comments:

The current study aims to investigate a proof-of-concept experiment of a new in-air beamline electron beam system to potentially degrade PFAS in aqueous and theoretically other matrices. It is extremely important to investigate innovative technologies to completely degrade PFAS and e-beam technology is one such approach. The authors have done a good job of summarizing existing literature on this field and accordingly comparing their proof-of concept device to these devices. This information is very relevant and is necessary to the PFAS community. This paper can be improved with more explanation and discussion of certain points, as suggested below. I recommend “Major revisions” to the manuscript at this stage.

We addressed the “Major Revisions” by systematically reviewing the manuscript for technical consistency, improving the cost analysis, refining explanations, and enhancing readability. Content changes were made where requested, and additional clarifications were added to strengthen the manuscript

Comments by line:

Line 16: Does not require citation, it is commonly known that it is a human right.

We removed the citation as suggested.

Lines 25-27: As a suggestion, please add a few sentences describing the current treatment technologies to remove and destroy PFAS and provide a more detailed pros and cons list of each of these technologies.

Lines 30–40 now describe granular activated carbon (GAC), ion exchange (IX), and reverse osmosis (RO) methods, with advantages (scalability, regulatory acceptance) and disadvantages (secondary waste generation, reduced efficiency for short-chain PFAS). Lines 40–45 contrast these with EBWT, highlighting its destructive, additive-free nature.

Lines 52-53: Which key PFAS? Please specify the PFAS and provide a list/grouping of the 20 PFAS mentioned.

Lines 66–71 now define PFAS-4 (PFOA, PFNA, PFHxS, PFOS) and PFAS-20 (20 specific C4–C13 perfluorocarboxylic and perfluorosulfonic acids) as per the EU Drinking Water Directive.

Line 61: Highly contaminated means what concentrations?

Lines 73–78 now explain that treated water is reintroduced into the system and filters are replaced about twice per year. Additional context (lines 78–86) contrasts our EBWT vision with current filtration methods.

Line 80, 130: Formatting error, please correct citation.

All formatting errors were corrected.

Line 103: Check units for the cost.

We reviewed all cost units and corrected inconsistencies throughout the manuscript. The cost analysis was expanded (lines 113–116, 130–132, 144–146, 150–151, 161–166) to detail accelerator types, capacities, and operating expenses.

Line 127: Define first usage- HZB, FWHM.

First mentions now include full definitions: Helmholtz-Zentrum Berlin (HZB) and full width at half maximum (FWHM).

Lines 191-195: Is it well accepted that higher dose rates means higher abundance of hydrated electrons? Is there a cutoff after which scavenging reactions dominate formation of hydrated electrons and their relative abundance?

Lines 244–248 now explain that higher dose rates do not necessarily yield more hydrated electrons due to scavenging species. Our study aims to identify the optimal dose rate.

Lines 212-213: Please reframe this statement, “to achieve a minimum dose of XX kGY".

Line 267 now specifies a minimum target dose of 10 kGy based on prior successful studies.

Lines 234-235: So how are the authors determining that the dose is high enough to degrade PFAS? Is this based on previous studies that have looked at PFAS degradation? Or is this based on model output? Does this actually consider the interactions between PFAS (C-F bonds) and hydrated electrons?

Lines 291–299 explain that our dose assumptions are based on previous studies achieving PFAS degradation in similar dose ranges. The planned experiment will verify this in practice.

Based on the operational parameters, what is the range of e-beam dose that can be delivered, and can the dose rate be tweaked?

Lines 436–439 state that our system can deliver dose rates up to several kGy/s; final average dose depends on setup parameters, which can be tuned experimentally.

Additional comments by the Editor

Double spacing was adjusted, the Fig7 was named correctly in the file upload. The affiliations were all corrected.

We have deposited all author-generated code underpinning our results in Zenodo, ensuring public access without restrictions. The repository contains all scripts, documentation, and example data needed to reproduce our analyses. It is archived under the MIT License, in line with PLOS ONE’s code-sharing guidelines. DOI: https://doi.org/10.5281/zenodo.16894583

3. Please amend your list of authors on the manuscript to ensure that each author is linked to an affiliation. Authors’ affiliations should reflect the institution where the work was done (if authors moved subsequently, you can also list the new affiliation stating “current affiliation:….” as necessary).’

We corrected the author list to ensure each author is linked to the correct affiliation at the time of the work.

4. Please ensure that you refer to Figure 7 in your text as, if accepted, production will need this reference to link the reader to the figure.

We now explicitly reference Figure 7 in the main text

5. We note you have included a table to which you do not refer in the text of your manuscript. Please ensure that you refer to Table 3, 4, and 5. in your text; if accepted, production will need this reference to link the reader to the Table.

Tables 3, 4, and 5 are now cited in the manuscript to ensure proper linking during production.

---

## [Decision Letter · Decision Letter 1]

17 Nov 2025

Compact High Power, Medium Energy Electron Accelerator for Treatment of Per- and Polyfluoroalkyl Contaminations in Water - EBWT@SEALAB

PONE-D-25-19239R1

Dear Dr. Spohr,

We’re pleased to inform you that your manuscript has been judged scientifically suitable for publication and will be formally accepted for publication once it meets all outstanding technical requirements.

Kind regards,

Alexandre Bonatto

Academic Editor

PLOS ONE

Additional Editor Comments (optional):

As I previously stated, the decision was taken based based on a single reviewer's report, who is a specialist on the paper's subject.

Reviewers' comments:

Reviewer's Responses to Questions

**Comments to the Author**

1. If the authors have adequately addressed your comments raised in a previous round of review and you feel that this manuscript is now acceptable for publication, you may indicate that here to bypass the “Comments to the Author” section, enter your conflict of interest statement in the “Confidential to Editor” section, and submit your "Accept" recommendation.

Reviewer #1: All comments have been addressed

2. Is the manuscript technically sound, and do the data support the conclusions?

Reviewer #1: Yes

3. Has the statistical analysis been performed appropriately and rigorously? 

Reviewer #1: Yes

4. Have the authors made all data underlying the findings in their manuscript fully available?

Reviewer #1: (No Response)

5. Is the manuscript presented in an intelligible fashion and written in standard English?

Reviewer #1: Yes

6. Review Comments to the Author

Reviewer #1: I have re-reviewed the manuscript and my comments have been addressed. The authors have done a great job at responding to the comments and adding more details to strengthen the manuscript.

7. PLOS authors have the option to publish the peer review history of their article (what does this mean?). If published, this will include your full peer review and any attached files.

Reviewer #1: **Yes: **Dr. Kaushik V Londhe

---

## [Editor Report · Acceptance letter]

PONE-D-25-19239R1

PLOS ONE

Dear Dr. Spohr,

I'm pleased to inform you that your manuscript has been deemed suitable for publication in PLOS ONE. Congratulations! Your manuscript is now being handed over to our production team.

Kind regards,

on behalf of

Dr. Alexandre Bonatto

Academic Editor

PLOS ONE